Abundance-weighted phylogenetic diversity measures distinguish microbial community states and are robust to sampling depth

McCoy Connor O.
Matsen Frederick A. IV matsen@fhcrc.org
Fred Hutchinson Cancer Research Center , Seattle, WA , United States
Chistoserdova Ludmila
Electronic publication date: 2013 Sep 12
Publication date: 2013
Volume: 1
Electronic Location ID: e157
Received 2013 May 1; Accepted 2013 Aug 20
Copyright: © 2013 McCoy et al.
Copyright year: 2013
Copyright holder: McCoy et al.
License: This is an open access article distributed under the terms of the Creative Commons Attribution License, which permits unrestricted use, distribution, and reproduction in any medium, provided the original author and source are credited.
License URL: https://creativecommons.org/licenses/by/3.0/

Keywords: Alpha diversity, Diversity indices, Phylogenetic diversity, Microbial ecology, Human microbiome

Funding: NIH R01 AI038518 Fred Hutchinson Cancer Research Center This work was supported by NIH R01 AI038518 and startup funds from the Fred Hutchinson Cancer Research Center. The funders had no role in study design, data collection and analysis, decision to publish, or preparation of the manuscript.

==============================
In microbial ecology studies, the most commonly used ways of investigating alpha (within-sample) diversity are either to apply non-phylogenetic measures such as Simpson’s index to Operational Taxonomic Unit (OTU) groupings, or to use classical phylogenetic diversity (PD), which is not abundance-weighted. Although alpha diversity measures that use abundance information in a phylogenetic framework do exist, they are not widely used within the microbial ecology community. The performance of abundance-weighted phylogenetic diversity measures compared to classical discrete measures has not been explored, and the behavior of these measures under rarefaction (sub-sampling) is not yet clear. In this paper we compare the ability of various alpha diversity measures to distinguish between different community states in the human microbiome for three different datasets. We also present and compare a novel one-parameter family of alpha diversity measures, BWPDθ, that interpolates between classical phylogenetic diversity (PD) and an abundance-weighted extension of PD. Additionally, we examine the sensitivity of these phylogenetic diversity measures to sampling, via computational experiments and by deriving a closed form solution for the expectation of phylogenetic quadratic entropy under re-sampling. On the three datasets, a phylogenetic measure always performed best, and two abundance-weighted phylogenetic diversity measures were the only measures ranking in the top four across all datasets. OTU-based measures, on the other hand, are less effective in distinguishing community types. In addition, abundance-weighted phylogenetic diversity measures are less sensitive to differing sampling intensity than their unweighted counterparts. Based on these results we encourage the use of abundance-weighted phylogenetic diversity measures, especially for cases such as microbial ecology where species delimitation is difficult.

Introduction

It is now well accepted that incorporating phylogenetic information into alpha (single-sample) and beta (between-sample) diversity measures can be useful in a variety of ecological contexts. Phylogenetic equivalents of all of major alpha diversity measures have been developed (Table 1). Starting with Faith’s original definition of phylogenetic diversity (Faith, 1992), which generalizes species count, there are now phylogenetic generalizations of the Simpson index to Rao’s quadratic entropy (Rao, 1982; Warwick & Clarke, 1995), the Shannon index to phylogenetic entropy (Allen, Kon & Bar-Yam, 2009), and the Hill numbers to qD(T) (Chao, Chiu & Jost, 2010). Phylogenetic diversity itself has been extended to incorporate taxon counts (Barker, 2002) and proportional abundance (Vellend et al., 2011). There have also been abundance-weighted measures that explicitly measure phylogenetic community structure (Fine & Kembel, 2011), or an “effective number of species” (Chao, Chiu & Jost, 2010). Many diversity measures can be tidily expressed in the framework of Leinster & Cobbold (2012), although the expression of phylogenetic diversity measures for non-ultrametric trees is complex.

Table 1 Overview of phylogenetic diversity measures used in the text.

phylogenetic diversity (Faith, 1992)	
phylogenetic generalization of species count	
phylogenetic quadratic entropy (Rao, 1982; Warwick & Clarke, 1995)	
phylogenetic generalization of the Simpson index	
phylogenetic entropy (Allen, Kon & Bar-Yam, 2009)	
phylogenetic generalization of the Shannon index	
qD(T) (Chao, Chiu & Jost, 2010)	
phylogenetic generalization of Hill numbers	
BWPD1 (Barker, 2002; Vellend et al., 2011)	
abundance-weighted version of phylogenetic diversity	
BWPDθ (this paper)	
one-parameter family interpolating between PD and BWPD1	

In this paper we use three example human microbiome datasets to demonstrate the utility of abundance-weighted phylogenetic diversity measures. We also introduce a one-parameter family interpolating between classical PD and an abundance-weighted generalization. We call the parameter θ and denote the one-parameter family BWPDθ; BWPD0 is classical PD, whereas BWPD1 is balance-weighted phylogenetic diversity, effectively PDaw of Vellend et al. (2011). Intermediate values of θ allow a partially-abundance-weighted compromise. Such a compromise has recently been shown to be useful for measuring beta diversity, with the introduction of a one-parameter family of “generalized UniFrac” measures (Chen et al., 2012). We use the name Balance Weighted Phylogenetic Diversity as described below because there are a variety of abundance weighted phylogenetic diversity measures. We compare the behavior of PD measures, including BWPDθ, under various levels of sampling using theory and example datasets.

Materials and Methods

Datasets

We apply the methods described below to three previously described 16S rRNA surveys of the human microbiome. The first two datasets are composed of samples from “normal” and dysbiotic microbial communities, where previous studies have associated changes in diversity with changes in health. The third dataset investigates the changes of the skin microbiome through time.

Bacterial vaginosis

First, we reanalyze a pyrosequencing dataset describing bacterial communities from women being monitored in a sexually transmitted disease clinic for bacterial vaginosis (BV). BV has previously been shown to be associated with increased community diversity (Fredricks, Fiedler & Marrazzo, 2005). For this study, swabs were taken from 242 women from the Public Health, Seattle and King County Sexually Transmitted Diseases Clinic between September 2006 and June 2010 of which 220 samples resulted in enough material to analyze (Srinivasan et al., 2012).

Selection of reference sequences and sequence preprocessing were performed using the methods described in Srinivasan et al. (2012). 452,358 reads passed quality filtering, with a median of 1,779 reads per sample (range: 523–2,366).

Oral periodontitis

We also utilize sequence data from a study of subgingival communities in 29 subjects with periodontitis, along with an equal number of healthy controls (Griffen et al., 2011a). The publication analyzing this dataset showed increased community diversity in samples from dysbiotic patients compared to healthy controls. Raw sequences were filtered, retaining only those reads with: a mean quality score of at least 25, no ambiguous bases, at least 150 base pairs in length, and an exact match to the sequencing primer and barcode. A total of 759,423 reads passed quality filtering, with a median of 8,320 reads per sample (range: 4,096–14,319).

As the phylogenetic placement method used below to calculate our measures requires a reference tree and alignment, we created a tree with FastTree 2.1.4 (Price, Dehal & Arkin, 2010) using the alignment and accompanying taxonomic annotation from the curated CORE database of oral microbiota (Griffen et al., 2011b).

Skin microbiome through time

Our third dataset is a study of skin microbial diversity through adolescence (Oh et al., 2012). Aligned sequences were obtained courtesy of the authors, although sequence data is available under the accession numbers [GQ000001] to [GQ116391] and can be accessed through BioProject ID 46333. A total of 90,142 Sanger sequences were available, with a median of 693 sequences per sample (range: 317–2884).

Balance-weighted phylogenetic diversity

In this section we introduce BWPDθ, our one-parameter family interpolating between classical PD and fully balance-weighted phylogenetic diversity. We will primarily consider so-called unrooted (Pardi & Goldman, 2007) phylogenetic diversity, which does not necessarily include the root. The case of rooted phylogenetic diversity can be calculated in a similar though simpler way as described below. Although we will primarily be working in an unrooted sense, it will be useful to use terminology that corresponds to the rooted case. For this reason, if the tree is not already rooted, assume an arbitrary root has been chosen; let the proximal side of a given edge be the side that contains the root and distal be the other.

We will describe BWPDθ in terms of a phylogenetic tree T with leaves L, and a contingency table describing the number of observations of the organisms at the leaves in various samples. The contingency table has rows labeled with the leaves of T, and columns labeled by samples. In microbial ecology this is frequently known as an OTU table. The entry corresponding to a given leaf and a given sample is the number of times that leaf was observed in that sample.

The classical (unrooted) phylogenetic diversity of a given sample in this context is the total branch length of the tree subtended by the leaves in that sample.

The path to generalizing PD is to note that this can be expressed as a sum of branch lengths multiplied by a step function. Let f(x) be the function that is one for x > 0 and zero otherwise. Let g(x) = min(f(x), f(1−x)) and Ds(i) be the fraction of reads in sample s that are in leaves on the distal side of edge i. Phylogenetic diversity can be then expressed as (1) PDu(s)=∑iℓig(Ds(i)).

That is, the sum of edge lengths in T which have reads from s on both the distal and proximal side.

Figure 1 gθ curves for various θ parameters.

As θ goes to zero, the gθ converge pointwise to g, which is 1 on the interior of the unit interval and 0 on the boundaries.

Note that the step function g is the limit of a one-parameter family of functions (Fig. 1). Indeed, defining (2) gθ(x)=2min(x,1−x)θ,

g is the pointwise limit of the gθ on the closed unit interval as θ goes to zero. Thus our one-parameter generalization is (3) BWPDθ(s)=∑iℓigθ(Ds(i)).

Note that when θ = 0 this is PD and when θ = 1 this is an abundance-weighted version of PD equivalent to executing the ΔnPD recipe of Barker (2002) up to a multiplicative factor.

The rooted equivalent of (3) is (4) RBWPDθ(s)=∑iℓi(Ds(i))θ,

which interpolates between rooted PD and an abundance-weighted version. Vellend et al. (2011) describe a measure, PDaw, which is equal to RBWPD1 multiplied by the total number of branches in T.

We call BWPD1 balance-weighted phylogenetic diversity because it weights edges according to the balance of read fractions on either side of an edge–edges with even amount of mass on either side are up-weighted, while edges with an uneven balance of mass are down-weighted. Indeed, if |x−(1−x)| is thought of as the imbalance of read fraction on either side of an edge, then 1−|x−(1−x)| is a measure of balance; note that on the unit interval, 2min(x, 1−x) = 1−|x−(1−x)|. Because a small x or an x close to 1 gives a small coefficient in the summation, small collections of reads or small perturbations of the read distribution will not change the value of BWPD1 appreciably.

Calculation of PD measures in example applications

Reads from the vaginal and oral studies were placed on a tree created from a curated set of taxonomically annotated reference sequences. As phylogenetic entropy and qD(T) operate on a rooted phylogeny, reference trees were assigned a root taxonomically (Matsen & Gallagher, 2012) meaning that a root was found that best separated high-level taxonomic groupings. pplacer was run in posterior probability mode (using the -p and --informative-prior flags), which defines an informative prior for pendant branch lengths with a mean derived from the average distances from the edge in question to the leaves of the tree. The resulting set of placements were classified at the family rank using a hybrid classifier implemented in the guppy tool from the pplacer suite. The hybrid classifier assigns taxonomic annotations to sequences using the combination of a naïve Bayes classifier (Wang et al., 2007) with a phylogenetic classifier (N Hoffman, A Gallagher and F Matsen, unpublished data). Any reads that could not be confidently classified to the family rank were not used in measures based on classification.

Full-length 16S sequences were available for the skin data, and so a more traditional tree-building approach was used. Representative OTUs were chosen for each site by clustering at 97% identity using USEARCH 5.1 (Edgar, 2010), with trees built on OTU centroids using FastTree (Price, Dehal & Arkin, 2010). To conform with methods used in Oh et al. (2012), the naïve Bayes classifier (Wang et al., 2007) was used to infer genus-level classifications to taxonomically root the tree; in our case we used the RDP classifier v2.5. The contingency (OTU) tables generated by clustering were made available to our tools via the BIOM (McDonald et al., 2012) format.

PDu (unrooted PD), phylogenetic quadratic entropy (Rao, 1982), phylogenetic entropy (Allen, Kon & Bar-Yam, 2009), and qD(T) (Chao, Chiu & Jost, 2010) were implemented for phylogenetic placements in the freely-available pplacer suite of tools (Matsen, Kodner & Armbrust, 2010) (http://matsen.fhcrc.org/pplacer) in the subcommand guppy fpd. Prior to diversity estimation, either phylogenetic placements were rarefied to the read count of the specimen in the dataset with the fewest sequences using guppy rarefy, or the corresponding rarefaction on full-length sequences was performed with the QIIME (Caporaso et al., 2010) rarefaction tool single_rarefaction.py. The mean value of each statistic over 100 such rarefactions was used for analysis.

Discrete measures of alpha diversity and richness were calculated on contingency tables obtained from clustering and taxonomic classification. Sequences were clustered into Operational Taxonomic Units (OTUs) at a 97% identity threshold using USEARCH 5.1 (Edgar, 2010). Similar results were observed when clustering at 95% identity (results not shown). OTU counts and family-level taxon counts were then rarefied as above in R 3.0.1 (R Development Core Team, 2013) using the vegan package (Oksanen et al., 2012). We obtained values for the Simpson (1949) and Shannon (1948) diversity indices, as well as the Chao1 (Chao, 1984) and ACE (Chao & Lee, 1992) measures of species richness using vegan functions diversity and estimateR.

Comparative analysis of alpha diversity measures

To investigate the relation between various measures of alpha diversity, we calculated Pearson’s r between all pairs of measures using the function rcorr from the R package Hmisc (Harrell, 2012). We then performed hierarchical clustering with the R function hclust, using d = 1−r as the distance between two measures.

Association of each measure with clinical criteria for the first two datasets was evaluated by examining the accuracy of a logistic regression using the measure as the sole predictor of whether the sample came from a “normal” or dysbiotic subject. In the vaginal dataset, we assessed each measure’s ability to predict whether a sample was from a subject positive for BV by Amsel’s criteria, a clinical diagnostic method (Amsel et al., 1983). In the oral dataset, we assessed each measure’s ability to predict whether a sample was from a healthy control, or a subject with periodontitis. Accuracy in predicting sample community state was assessed by leave-one-out cross-validation using the R package boot (Davison & Hinkley, 1997; Canty & Ripley, 2012).

For the vaginal dataset, we also calculated R2 values using each measure individually as a predictor for sample Nugent score in a linear regression. The Nugent score provides a diagnostic score for BV which ranges from 0 (BV-negative) to 10 (BV-positive) based on presence and absence of bacterial morphotypes as viewed under a microscope (Nugent, Krohn & Hillier, 1991).

We calculated p-values to compare within- and between-stratification variability using R’s built-in t.test function for the vaginal data, which had a binary stratification, and the aov function for the oral and skin datasets. The vaginal dataset was stratified by Amsel’s criterion, the oral dataset by condition and sampling site, and the skin microbiome dataset by Tanner scale of physical development (Oh et al., 2012). Note that we are not presenting these uncorrected p-values as evidence that there is an interesting relationship between the microbiome and a given stratification, but rather are using p-values as a way of measuring within-stratum heterogeneity compared to between-stratum heterogeneity for the various measures.

Results

Application to the human microbiome

Vaginal microbiome

Like Srinivasan et al. (2012) and many others in the field, we observe greater diversity in BV positive specimens using a variety of diversity and richness measures (Fig. S1). In particular, this is true for BWPDθ for a variety of values of θ (Fig. S2).

Table 2 Correlation and predictive performance of the various alpha diversity measures applied to the vaginal dataset.

Rows are ordered by increasing mean rank across performance measurements. Nugent R2:R2 value using the measure as a predictor, and the Nugent score as response in a linear model. Amsel accuracy: proportion of specimens with correct BV classification under a leave-one-out cross-validation. Amsel p-value: p-value from a two-sample t-test on values stratified by BV classification. “OTU” designates the measure applied to 97% clustering groups, and “Family” designates taxonomic classification at the family level. Measures described in main text.

Measure	Amsel accuracy	Nugent R2	Amsel p-value	Mean rank	
PDu	0.834	0.737	1.84E−35	1.3	
BWPD0.25	0.836	0.735	1.98E−35	2.0	
Simpson (Family)	0.817	0.735	4.11E−33	4.0	
BWPD0.5	0.827	0.700	3.33E−33	4.3	
Shannon (Family)	0.813	0.724	2.28E−32	5.3	
Phylo. entropy	0.831	0.679	1.81E−31	5.7	
Chao1 (Family)	0.813	0.704	6.27E−31	7.0	
0.5D(T)	0.818	0.658	7.47E−29	8.0	
0.25D(T)	0.809	0.682	2.25E−30	8.3	
Phylo. quad. entropy	0.813	0.648	7.89E−30	9.0	
BWPD1	0.795	0.611	5.38E−28	11.0	
Chao1 (OTU)	0.766	0.488	1.64E−23	12.7	
ACE (Family)	0.766	0.491	2.82E−11	13.0	
ACE (OTU)	0.764	0.469	6.82E−22	13.7	
Shannon (OTU)	0.758	0.380	5.27E−16	14.7	
Simpson (OTU)	0.697	0.191	1.42E−07	16.0	

In the vaginal data, phylogenetic measures of alpha diversity have better cross-validation accuracy for the Amsel classification and better correlation with the Nugent score than discrete OTU-based measures (Table 2). All measures were somewhat accurate in identifying community state, with even the worst performers classifying almost 70% of samples correctly. BWPD0.25, BWPD0.5, PDu, and phylogenetic entropy perform well predicting BV status. Correlation with Nugent score varies from 0.19 using Simpson (OTU) to 0.74 using PDu. OTU-based measures rank in the bottom half of the measures tested, and below all phylogenetic measures.

Figure 2 Dendrogram relating alpha diversity measures applied to the vaginal dataset.

In the hierarchical clustering of alpha measures on the vaginal dataset, phylogenetic methods are separated from OTU-based methods (Fig. 2). BWPDθ is similar to different extant phylogenetic alpha diversity measures for different θ. The Simpson and Shannon diversity measures cluster together, as do the ACE and Chao1 richness measures.

Figure 3 Comparison of rarefied and unrarefied values of various phylogenetic alpha diversity measures as applied to the vaginal dataset.

The value of six alpha measures for each specimen using all available sequences is plotted on the x-axis. The value of the alpha measures for each specimen after a single rarefaction to 523 sequences (the smallest sequence count across specimens) is plotted on the y-axis. The y = x line is shown in blue.

Figure 3 shows values of BWPDθ calculated before (x-axis) and after (y-axis) a single rarefaction to 523 sequences per sample. Samples for which the BWPDθ value changes little lie close to the blue line, which shows the case of no difference between original and rarefied samples. Increasing θ, which corresponds to increased use of abundance information, reduces the change in BWPDθ induced by rarefaction. Phylogenetic quadratic entropy and phylogenetic entropy both show behavior similar to BWPD1, with rarefaction introducing little effect.

It might be possible to formalize a statement to this effect by computing the expectation of these alpha measures under rarefaction. However, computing the expectation for BWPDθ under rarefaction does not appear to be straightforward: the methods of Dremin (1994) might be applicable in this setting, however, even the integer moments of the hypergeometric distribution are complicated and the non-integer moments are bound to be very complex. We have, however, shown in the Appendix that the expectation of phylogenetic quadratic entropy under rarefaction to k sequences assigned to the tips of a phylogenetic tree is E[PQEk]=k−1kn(n−1)∑iℓidi(n−di)

where di is the number of sequences falling below edge i and ℓi is the length of edge i. This is almost identical to the unrarefied value of phylogenetic quadratic entropy, i.e. PQE=1n2∑iℓidi(n−di).

Thus it is not surprising to see that the expectation of PQE under rarefaction is very close to the original value (Fig. S3) for reasonably large k and n.

Oral microbiome

As previously observed by Griffen et al. (2011a), we find generally higher diversity in samples from dysbiotic patients (Fig. 4). We evaluated the ability of each alpha diversity measure to predict whether a sample came from an individual with periodontitis, regardless of sample collection site, using the above methods.

In the oral dataset, phylogenetic alpha diversity measures incorporating abundance gave the best predictions of community state (Table 3, Fig. 4). In contrast, classical phylogenetic diversity performed less well. These results were almost identical in terms of rank order after applying additional quality filtering steps to correct sequencing errors and remove potentially chimeric sequences (Table S1).

Table 3 Predictive accuracy of each measure in the oral dataset and p-value from an ANOVA stratified by disease status and sampling site.

Measure	Diseased status accuracy	ANOVA p-value	Mean rank	
Phylo. entropy	0.791	4.97E−09	1.0	
BWPD0.5	0.782	6.50E−09	2.0	
BWPD0.25	0.755	7.16E−08	4.0	
Phylo. quad. entropy	0.770	2.47E−07	4.0	
Simpson (Family)	0.776	1.45E−06	4.0	
0.5D(T)	0.734	4.74E−06	6.5	
0.25D(T)	0.735	4.33E−05	7.0	
PDu	0.691	6.37E−06	8.5	
Shannon (Family)	0.734	5.32E−05	8.5	
BWPD1	0.698	3.57E−04	9.5	
Chao1 (OTU)	0.685	9.94E−04	11.0	
ACE (OTU)	0.682	1.30E−03	12.0	
Simpson (OTU)	0.676	2.39E−02	13.5	
Shannon (OTU)	0.672	1.31E−03	14.0	
Chao1 (Family)	0.674	2.64E−01	15.0	
ACE (Family)	0.663	1.82E−01	15.5	

OTU-based methods and phylogenetic methods are not as separated in a hierarchical clustering as for the vaginal dataset (Fig. S6). However, many of the same pairings are present in both clusterings: BWPD0.5 with phylogenetic entropy, BWPD1 with quadratic entropy, Simpson with Shannon, and ACE with Chao1. Interestingly, PDu, BWPD0.25, and the qD(T) measures all cluster with the discrete richness measures ACE and Chao1.

Like the vaginal dataset, incorporating abundance information decreases the effect of rarefaction on BWPDθ values (Figs. S4 and S5).

Figure 4 Comparison of diversity between samples from healthy controls, healthy sites of dysbiotic patients, and dysbiotic sites of dysbiotic patients on the oral dataset, using different measures of alpha diversity.

“Shallow” means a shallow pocket between tooth and gum tissue, while “deep” means a sample from a deep pocket between gum tissue that has separated from its tooth. Top row: cluster-based methods. Bottom rows: phylogenetic methods.

Skin microbiome

To further assess resolution and robustness of phylogenetic diversity measures, we considered skin microbiome data from a study by Oh et al. (2012). This study tracked the changes of the skin microbiome through “Tanner” developmental stages of adolescence (Tanner & Whitehouse, 1976). Because there are five Tanner stages, and they do not have a monotonic relationship with skin microbiome diversity (Oh et al., 2012), we focused on ANOVA p-values to see if the diversity measurements had small within-stage heterogeneity compared to between-stage heterogeneity. To compare the ANOVA p-values associated with the diversity measurements across the various datasets, we ranked the p-value of the diversity measures from lowest to highest for each dataset individually. We averaged these ranks to gain an overall measure of performance. The results again show phylogenetic measures generally performing better than OTU-based measures (Table 4). This finding holds true even after removing potential chimeras (Table S1). In this case, a light weighting or no weighting of phylogenetic diversity by abundance performed better than full abundance-weighting.

Table 4 ANOVA p-values for various diversity statistics applied to the skin microbiome data of Oh et al. (2012).

Rows are ordered by increasing mean rank across sites.

	Ac	N	Pc	Vf	Mean rank	
PDu	3.34e−02	4.94e−03	3.03e−03	2.28e−04	4.50	
BWPD0.25	4.22e−02	1.32e−03	6.37e−03	5.86e−04	5.25	
BWPD0.5	8.54e−02	9.85e−05	3.65e−02	5.85e−03	6.00	
Shannon (OTU)	6.61e−02	9.48e−02	9.65e−02	1.05e−05	6.50	
Chao1 (OTU)	8.00e−02	6.46e−03	3.98e−03	3.18e−03	6.75	
Phylo. quad. entropy	2.52e−01	1.12e−05	4.99e−01	1.67e−01	7.50	
Phylo. entropy	1.37e−01	1.15e−03	1.55e−01	2.09e−02	7.75	
0.5D(T)	8.91e−01	5.63e−04	3.84e−03	9.09e−01	8.25	
0.25D(T)	7.00e−01	2.27e−03	2.35e−03	9.41e−01	8.50	
BWPD1	3.09e−01	5.95e−05	6.65e−01	5.41e−01	8.50	
0D(T)	4.42e−01	1.05e−02	1.38e−03	9.38e−01	8.75	
Simpson (OTU)	9.38e−02	4.01e−01	8.49e−01	1.01e−04	8.75	
Notes.

The same site abbreviations are used as in their paper

Ac antecubital fossa

N nares

Pc popliteal fossa

Vf volar forearm

Applications summary

In all three of the datasets investigated, abundance-weighted phylogenetic diversity measures showed good performance to distinguish between community states: between “normal” and dysbiotic samples in the oral and vaginal microbiomes, and between developmental stages in the skin microbiome. Notably, the best distinguishing measure in each dataset was phylogenetic; in addition BWPD0.25 and BWPD0.5 were the only measures that were in the top four for all datasets. The result that partial abundance weighting performs well corresponds to analogous results for beta diversity, where an intermediate exponent for “generalized UniFrac” was the most powerful (Chen et al., 2012).

Discussion

Phylogenetic alpha diversity measures were more closely related to community state than were discrete measures based on OTU clustering for the datasets investigated here. This result is especially interesting given that the Simpson index, the Shannon index, or counting applied to OTU tables are very common ways of characterizing microbial diversity (Fierer et al., 2007; Grice et al., 2009; Hill et al., 2003; Dethlefsen & Relman, 2011). As also noted by Aagaard et al. (2012), we find that measurements of diversity using taxonomic classification can be useful in describing communities, and in fact perform much better than the same measurements of diversity applied to OTU counts; however, this approach requires a taxonomically well characterized environment. Our results can be viewed as an experimental confirmation of the notion that incorporating similarity between species is important to get sensible measures of diversity, which has been advocated by many, including most recently by Leinster & Cobbold (2012).

We find that classical phylogenetic diversity is sensitive to sampling depth, underestimating the true value in small samples. Biases have also been described for diversity measures using OTU tables (Gihring, Green & Schadt, 2012). In contrast, we observe that some abundance-weighted phylogenetic measures are relatively robust to varying levels of sampling.

These results did not appear to be the result of sequencing issues. In principle, OTU methods could have performed badly because of error-prone and chimeric sequences inflating the number of OTUs. Although this is a real danger for OTU quantification, in this study its impact appears to be limited–similar results were obtained with the oral data after de-noising and chimera removal and the skin data (which used Sanger sequencing) after chimera removal.

We note that on our data, non-phylogenetic measures applied to family level taxonomic groupings are generally more discriminating than the corresponding measures applied to OTUs. This may be because our sequences are from the human microbiome, and taxonomic classification is especially well-developed in that setting; in particular the taxonomic names may already be defined in a way that corresponds to dysbiosis. Thus this particular difference may not continue to be true in a taxonomically less-well-characterized environment.

As of the publication of this paper, no abundance-weighted phylogenetic alpha diversity measures are implemented in either mothur (Schloss et al., 2009) or QIIME (Caporaso et al., 2010), two of the most popular tools for analysis of microbial ecology data. Although the fact that abundance-weighted phylogenetic diversity measures performed very well for the three datasets investigated here does not imply that they are best in general, we suggest that abundance-weighted phylogenetic measures be given greater consideration for microbial ecology studies. For this to happen, implementations in commonly used microbial ecology software packages will be needed, in addition to our implementation and that of the picante R package (Kembel et al., 2010).

Supplemental Information

Supplemental Information 1 Supplementary mathematical results, Tables, and Figures

PDF containing a derivation and a more general description of weighted PD, as well as a number of supplementary figures and tables.

Click here for additional data file.

The authors would like to thank Steven Kembel for encouragement and guidance, Steven N. Evans for probability consultation, and David Nipperess for an interesting dialog concerning phylogenetic diversity and rarefaction. The Segre lab at the NIH, in particular Sean Conlan and Julia Oh, were very generous and helpful with the skin data. This work would not have been possible without an ongoing collaboration with David Fredricks, Noah Hoffman, Martin Morgan, and Sujatha Srinivasan at the Fred Hutchinson Cancer Research Center.

Additional Information and Declarations

Competing Interests

Author Contributions

The authors declare no competing interests.

Connor O. McCoy and Frederick A. Matsen IV conceived and designed the experiments, performed the experiments, analyzed the data, wrote the paper.

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
