# Peer review of "Abundance-weighted phylogenetic diversity measures distinguish microbial community states and are robust to sampling depth"

_PeerJ, doi:10.7717/peerj.157_

## Round 0.1 · original submission · Minor Revisions

· Academic Editor

Minor Revisions

I look forward to the revised manuscript.

·

Basic reporting

The submitted manuscript appears to adhere to all PeerJ policies and is written in very in a very clear and concise English. The Introduction provides a clear overview and presents the scope of the manuscript in relation to previous work.

Experimental design

In this manuscript, McCoy and Matsen provide a well-motivated measure of phylogenetic alpha-diversity (PD), able to take into interpolate between abundance-weighted (Barker 2002, as interpreted by Vellend) and classic -unweigheted PD . Further, they convincingly demonstrate that this measure is able to distinguish between different communities using a selection of three microbiome datasets from healthy and diseased individuals. The work is meaningful, rigorous and the methods described clearly.

However, filtering, clustering and other methods to remove PCR- and sequencing-induced artefacts were not addressed properly by the original authors in the two larger sequence datasets (vaginal and oral microbiomes). The first of these used filtering based only on quality scores and the second the RDP pipeline. It has been demonstrated that this method is not sufficient to remove such artefacts (see e.g. Quince et al 2011 (BMC Bioinformatics 12:38.), Schloss et al 2011 (PLoS ONE 6:e27310), Kunin et al 2009 (Environ Microbiol 12:118-123) and will result in inflated OTU richness estimates and skewed OTU-abundance relationships. PD should be less sensitive to such artefacts and this can very well be the main reason for why the estimate suggested here performed better than OTU-based ones. I think that this fact deserves to be mentioned in the Discussion. Better still, the authors could include a dataset in the comparison which has been handled using e.g. AmpliconNoise or DeNoiser and where chimeric sequences were properly addressed, or re-analyse the raw sequence data using such methods, resulting in a new OTU table. Failing this, I think that this common pitfuall of OTU-based analysis deserves to be mentioned in the Discussion as the richness estimates in the original articles do appear very high.

If the authors are interested in repeating a test with noise-cleaned sequence data and a more robust OTU table, I happily volunteer to help out with data-treatment (sequence cleaning)

Validity of the findings

See Experimental design above. Apart from that no comments.

Additional comments

Specific comments:

In the abstract, Simpson diversity is called a "count-only" measure. This is unclear and leads this reader to think more of OTU richness. In Table 1, Af and Pf are mentioned in the caption but in the headers "Ac" and "Pc" are used.

·

Basic reporting

meets standards

Experimental design

good

Validity of the findings

good

Additional comments

In this paper, McCoy and Matsen introduce a novel family of phylogenetic alpha diversity measures that interpolates between classical phylogenetic diversity (PD), which does not account for the abundance of phylogenetic lineages in a sample, and an abundance-weighted extension of PD. Using 3 published studies of the human gut microbiota, they evaluate how their new measure and other alpha diversity measures compare in their ability to differentiate samples in different categories, including healthy versus bacterial vaginosis, periodontitis and controls, and skin samples at different developmental stages. Overall, this paper highlights that phylogenetic alpha diversity measures can be more powerful than commonly used “discrete” measures that rely on OTU assignments. It also highlights that abundance-weighted phylogenetic alpha diversity measures can be more powerful than phylogenetic alpha diversity measures that do not account for abundances (i.e. PD). The authors point out that abundance-weighted phylogenetic diversity measures are not used commonly in studies of the microbiota and not implemented in commonly used analysis tools such as QIIME and mother, and argue based on their results that they should be. Overall, I agree with them, and thus think that this paper is a valuable contribution to the field. I did, however, think that there were some ways in which the paper could be improved.

1) I was confused by the treatment of rarefaction in this paper. My understanding is that for e.g. the data presented in Table 1, for the OTU and family based “discrete” measures (i.e. Shannon, Simpson, Chao1, and ACE), they rarefied the data, and then for the phylogenetic measures they did not rarefy, with the exception of PD, for which they present both the rarefied and unrarefied results. Why do it this way (i.e. rarefy one class of measures and not the other and then for one (PD) do it both ways?).

In general, I think that you should always rarefy. The variability in the number of sequences per sample has no real meaning (i.e. just an artifact of sequencing since equal amounts of DNA from each sample is added to the sequencer), and this variability has the potential to affect alpha diversity estimates. I realize that later in the paper (in Figure 3 and mathematically) they show that abundance weighted measures are not very sensitive to sampling depth, although I am not convinced that there will not be more sensitivity for environments that are very undersampled.

So, by making the point that abundance-weighted measures are not sensitive to sampling depth and then not rarefying the data in their analyses, are they trying to say that because of this lack of sensitivity that we should not rarefy when using these measures? What would be the advantage? The only thing that I can think is that you can potentially more accurately estimate alpha diversity with more data, but as this has not been demonstrated here, I still think that it is good practice to rarefy.

With regard to this, in Figure 3 and Fig. S4, they show that 0.25D(T) and BWPD0.25 are sensitive to sampling depth, and yet as far as I can tell they still do not used rarefied data in their analyses of all three of the microbiota datasets with these measures. On page 10, lines 276-278, the authors note “classical phylogenetic diversity was among the worst predictors; rarefaction did help…”. Why look at unrarefied at all when they show in Figure 3 themselves that sequencing depth matters and conceptually it of course makes sense that if you have not sequenced fully, the more sequences you look at with PD the more diversity you will see.

For all three studies, I would really like to see all of the measurements made on rarefied data.

2) The authors evaluate many different alpha diversity measures in this paper. One thing that would be really helpful is a table that described and classifies them all. Perhaps a columns that 1) designated discrete versus phylogenetic 2) designate abundance-weighted/non/in-between, 3) show an equation where appropriate, 4) briefly describes the measure with words, and 5) shows the info in paragraph 1 of the introduction of which measures are phylogenetic “versions” of particular discrete measures and 6) gives a reference.

3) The information given on page 3 on the example datasets should have better consistency on the types of information provided for each one. The description of the skin microbiome is particularly sparse. Information provided for the other samples, such as the range of sequences per sample, how these sequences were generated, quality filtered etc. should be provided.

4) It is kinda interesting that the discrete measures applied at the family level often appear more powerful than those at the OTU level. Any ideas on why this may be the case?

Minor comments:
1) In the sentence in the abstract “In all three of the datasets considered, an abundance-weighted measure is the best differentiator between community states.”, the authors should say “a phylogenetic abundance-weighted measure is the best differentiator” as stated could be a discrete abundance-weighted measure which didn't do so good.
2) Lines 141, 159: I am not sure what the authors mean exactly when they say that they “assigned the root taxonomically”
3) The meaning of “shallow” and “deep” in Fig 4 are not defined anywhere.
4) typos/grammatical
a. line 79: Fix Oh et al reference formatting
b. line 304: should Tab. 5 be Table 2?

---

## Round 0.2 · accepted · Accept

· Academic Editor

Accept

I find modifications to the original manuscript satisfactory.

---

## Author Rebuttal · Round 0.2

## Reviewer 1 (Anders Lanzén)

> In this manuscript, McCoy and Matsen provide a well-motivated measure of phylogenetic alpha-diversity (PD), able to take into interpolate between abundance-weighted (Barker 2002, as interpreted by Vellend) and classic -unweigheted PD . Further, they convincingly demonstrate that this measure is able to distinguish between different communities using a selection of three microbiome datasets from healthy and diseased individuals. The work is meaningful, rigorous and the methods described clearly.

Thank you.

> However, filtering, clustering and other methods to remove PCR- and sequencing-induced artefacts were not addressed properly by the original authors in the two larger sequence datasets (vaginal and oral microbiomes). The first of these used filtering based only on quality scores and the second the RDP pipeline. It has been demonstrated that this method is not sufficient to remove such artefacts (see e.g. Quince et al 2011 (BMC Bioinformatics 12:38.), Schloss et al 2011 (PLoS ONE 6:e27310), Kunin et al 2009 (Environ Microbiol 12:118-123) and will result in inflated OTU richness estimates and skewed OTU-abundance relationships. PD should be less sensitive to such artefacts and this can very well be the main reason for why the estimate suggested here performed better than OTU-based ones. I think that this fact deserves to be mentioned in the Discussion. Better still, the authors could include a dataset in the comparison which has been handled using e.g. AmpliconNoise or DeNoiser and where chimeric sequences were properly addressed, or re-analyse the raw sequence data using such methods, resulting in a new OTU table. Failing this, I think that this common pitfuall of OTU-based analysis deserves to be mentioned in the Discussion as the richness estimates in the original articles do appear very high. If the authors are interested in repeating a test with noise-cleaned sequence data and a more robust OTU table, I happily volunteer to help out with data-treatment (sequence cleaning)

Thank you for the astute suggestion. We point out that for the skin data does not have these sequence quality issues and also has the top three metrics being phylogenetic metrics. Additionally, in response to your comment we have done chimera removal on the skin dataset, and de-noising and chimera removal on the oral data set, and have found only minor differences between the original and "cleaned" differences. This is now noted in the Results and Discussion sections.

> Specific comments:

> In the abstract, Simpson diversity is called a "count-only" measure. This is unclear and leads this reader to think more of OTU richness.

Ah, thanks. We have changed to "non-phylogenetic."

> In Table 1, Af and Pf are mentioned in the caption but in the headers "Ac" and "Pc" are used.

Nice catch, thanks.

## Reviewer 2 (Catherine Lozupone)

> In this paper, McCoy and Matsen introduce a novel family of phylogenetic alpha diversity measures that interpolates between classical phylogenetic diversity (PD), which does not account for the abundance of phylogenetic lineages in a sample, and an abundance-weighted extension of PD. Using 3 published studies of the human gut microbiota, they evaluate how their new measure and other alpha diversity measures compare in their ability to differentiate samples in different categories, including healthy versus bacterial vaginosis, periodontitis and controls, and skin samples at different developmental stages. Overall, this paper highlights that phylogenetic alpha diversity measures can be more powerful than commonly used "discrete" measures that rely on OTU assignments. It also highlights that abundance-weighted phylogenetic alpha diversity measures can be more powerful than phylogenetic alpha diversity measures that do not account for abundances (i.e. PD). The authors point out that abundance-weighted phylogenetic diversity measures are not used commonly in studies of the microbiota and not implemented in commonly used analysis tools such as QIIME and mother, and argue based on their results that they should be. Overall, I agree with them, and thus think that this paper is a valuable contribution to the field. I did, however, think that there were some ways in which the paper could be improved.

Thank you.

> 1) I was confused by the treatment of rarefaction in this paper. My understanding is that for e.g. the data presented in Table 1, for the OTU and family based "discrete" measures (i.e. Shannon, Simpson, Chao1, and ACE), they rarefied the data, and then for the phylogenetic measures they did not rarefy, with the exception of PD, for which they present both the rarefied and unrarefied results. Why do it this way (i.e. rarefy one class of

measures and not the other and then for one (PD) do it both ways?).

In general, I think that you should always rarefy. The variability in the number of sequences per sample has no real meaning (i.e. just an artifact of sequencing since equal amounts of DNA from each sample is added to the sequencer), and this variability has the potential to affect alpha diversity estimates. I realize that later in the paper (in Figure 3 and mathematically) they show that abundance weighted measures are not very sensitive to sampling depth, although I am not convinced that there will not be more sensitivity for environments that are very undersampled.

So, by making the point that abundance-weighted measures are not sensitive to sampling depth and then not rarefying the data in their analyses, are they trying to say that because of this lack of sensitivity that we should not rarefy when using these measures? What would be the advantage? The only thing that I can think is that you can potentially more accurately estimate alpha diversity with more data, but as this has not been demonstrated here, I still think that it is good practice to rarefy.

With regard to this, in Figure 3 and Fig. S4, they show that 0.25D(T) and BWPD0.25 are sensitive to sampling depth, and yet as far as I can tell they still do not used rarefied data in their analyses of all three of the microbiota datasets with these measures. On page 10, lines 276-278, the authors note "classical phylogenetic diversity was among the worst predictors; rarefaction did help. . . ". Why look at unrarefied at all when they show in Figure 3 themselves that sequencing depth matters and conceptually it of course makes sense that if you have not sequenced fully, the more sequences you look at with PD the more diversity you will see.

For all three studies, I would really like to see all of the measurements made on rarefied data.

Thank you for the suggestion. We have changed to rarefied analyses across the board. This did have a substantial effect on some of the specifics, but our main observation still holds.

2) The authors evaluate many different alpha diversity measures in this paper. One thing that would be really helpful is a table that described and classifies them all. Perhaps a columns that 1) designated discrete versus phylogenetic 2) designate abundance-weighted/non/in-between, 3) show an equation where appropriate, 4) briefly describes the measure with words, and 5) shows the info in paragraph 1 of the introduction of which measures are phylogenetic "versions" of particular discrete measures and 6) gives a reference.

We have put in a table that we hope satisfies the spirit of this request. It lists all of the phylogenetic diversity measures used, gives a reference and gives a description. We decided to only put in the phylogenetic diversity measures as it seems fair to assume that the reader is familiar with the non-phylogenetic measures. Although we would have liked to add in equations, it would have taken several pages to fully explain and define all of the variables.

3) The information given on page 3 on the example datasets should have better consistency on the types of information provided for each one. The description of the skin microbiome is particularly sparse. Information provided for the other samples, such as the range of sequences per sample, how these sequences were generated, quality filtered etc. should be provided.

We have done this now, thanks.

4) It is kinda interesting that the discrete measures applied at the family level often appear more powerful than those at the OTU level. Any ideas on why this may be the case?

The second to last paragraph of the discussion is now dedicated to this.

1) In the sentence in the abstract "In all three of the datasets considered, an abundance-weighted measure is the best differentiator between community states.", the authors should say "a phylogenetic abundance-weighted measure is the best differentiator" as stated could be a discrete abundance-weighted measure which didn't do so good.

After rarefaction things did shift around a bit and so this sentence has been rewritten completely.

2) Lines 141, 159: I am not sure what the authors mean exactly when they say that they "assigned the root taxonomically"

This is developed in the referenced paper by Matsen and Gallagher, and a brief description has been added.

3) The meaning of "shallow" and "deep" in Fig 4 are not defined anywhere.

They are now defined in the figure legend, thanks.

4) typos/grammatical

    a. line 79: Fix Oh et al reference formatting

Thanks!

    b. line 304: should Tab. 5 be Table 2?

We have checked all table and figure references.